# The Effect of Cyclic Solution Heat Treatment on the Martensitic Phase Transformation and Grain Refinement of Co-Cr-Mo Dental Alloy

**Shahab Zangeneh [1,\*], Hamid Reza Lashgari [2,\*], Shaimaa Alsaadi [1], Sara Mohamad-Moradi [1] and Morteza Saghafi [3]**

[1]   Department of Materials and Textile Engineering, Faculty of Engineering, Razi University, Kermanshah 6714414971, Iran; shaimaa.alsa3dy@gmail.com (S.A.); saramohamadmoradi74@gmail.com (S.M.-M.)

[2]   School of Materials Science and Engineering, University of New South Wales, Sydney, NSW 2052, Australia

[3]   Department of Materials Science and Engineering, Faculty of Engineering, Imam Khomeini International University, Qazvin 3414896818, Iran; saghafiyazdi@gmail.com

\*   Correspondence: Shzangeneh@razi.ac.ir (S.Z.); h.lashgari@unsw.edu.au (H.R.L.); Tel.: +98-918-385-344-5 (S.Z.); +61-404-231-341 (H.R.L.)

**Abstract:** The present study was undertaken to investigate the effect of continuous and discontinuous (cyclic) solution heat treatment on the athermal and isothermal $\varepsilon$ martensite phase transformation in Co-28Cr-6Mo-0.3C implant alloy. The results showed that the cyclic solution heat treatment induced more of the athermal $\varepsilon$ martensite phase in the alloy than that of the continues one. In addition, the cyclic heat treatment contributes to the development of more isothermal martensite phase during isothermal aging at 850 °C and, moreover, grain refinement in the area beneath the sample surface. The severity of grain refinement was highly significant adjacent to the surface and decreased by increasing the distance from the sample free surface. This novel grain refinement in high-carbon Co-Cr-Mo alloy was attributed to the generation of larger quenching thermal stresses introduced beneath the surface during cyclic solution treatment. The repetitive heating/cooling cycle modifies the surface properties, refines the grain size and leads to uniform dispersion of the secondary carbides. The corrosion resistance of the cyclically solution heat-treated samples was superior as compared to the as-cast one.

**Keywords:** Co-Cr-Mo alloy; martensitic phase; solution treatment; grain refinement; reverse transformation

## 1. Introduction

Cast Co-Cr-Mo alloys were developed at the beginning of the 20th century with the addition of minor constituents such as carbon, nitrogen, etc. Cast components made of Co-Cr-Mo alloys have become very popular and have been widely used as gas turbine nozzles [1–4] and surgical implants (such as partial denture, screws in orthopedic for fracture-fixation, prosthetic joint replacements, and so on) due to their outstanding mechanical properties, wear resistance and, more importantly, good biocompatibility. However, long-term (lifetime) side effects of implantation due to brittleness, wear debris, metal ions and corrosion products continue to raise concerns because of the potential damage to human organs and tissues [5]. This may result in local tissue toxicity, cytotoxicity and possibly adverse genotoxicity effects. In general, most reported failures of different materials occur on the surface, such as fatigue fracture, wear and corrosion. Hence, the modification and optimization of the surface property such as hardness (related to wear resistance) and grain refinement could effectively reduce the

risk of implant material failure while in use in the human body. According to the classical Hall–Petch relationship (for grain sizes larger than 1 µm), the yield strength increases as grain size decreases which is attributed to the grain boundaries acting as obstacles to slip dislocations causing dislocation pile-up behind grain boundaries. Therefore, surface modification through grain refinement of the surface layer could effectively enhance the surface proportioned properties such as wear resistance (surface hardness) and fatigue performance [6,7].

Generally, in Co-based alloys, $\varepsilon$ phase with hexagonal close-packed (hcp) structure, are expected to be stable at room temperature, but $\gamma$ phase with a face-centered-cubic (fcc) structure is a typical and quite common structure at room temperature. In fact, the $\gamma_{fcc} \rightarrow \varepsilon_{hcp}$ phase transformation is quite sluggish under normal cooling conditions. Therefore, in Co-Cr-Mo alloys, the fcc structure is mostly retained at room temperature. Numerous works in the literature [8–10] have been devoted to investigating the isothermal $\gamma_{fcc} \rightarrow \varepsilon_{hcp}$ phase transformation which proceeds in two different stages. In each stage, different types of $\varepsilon$ phase (or hcp phase) with different morphologies denoted by $hcp_1$ and $hcp_2$ are formed. In the first stage, band-shaped $hcp_1$ phase with high density of stacking faults is formed on $(111)_\gamma$ planes and in the second one, pearlite-like $hcp_2$ phase (or lamellar phase) is formed which contains very few stacking faults. Pearlite-like structures are formed by the following reaction: $\gamma_{fcc} \rightarrow \varepsilon_{hcp} + M_{23}C_6$ carbides (M = Cr, Mo and Co).

Heat treatment processes including solution treatment at high temperature are employed to dissolve secondary phases and carbides followed by quenching and isothermal aging to tailor the microstructure and enhance the mechanical properties without deteriorating the biocompatibility and corrosion resistance. The effect of the solution heat treatment method, i.e., stationary heat treatment versus cyclic heat treatment, on the $\gamma_{fcc} \rightarrow \varepsilon_{hcp}$ phase transformation has been poorly reported while it could be important to those in the biomedical industry.

In general, the hot forging of high-carbon Co-Cr-Mo alloys could potentially lead to the crack formation during forging process impeding the fabrication of components through hot working and/or severe plastic deformation (SPD). It has been generally accepted that the SPD application is limited in Co-Cr-Mo alloys as a result of high work hardening through the strain-induced martensitic transformation. Moreover, generation of dislocation density during the SPD process causes to accelerate the release of metallic ions in Co-Cr-Mo and, in particular, that of Co. Therefore, grain refinement to reach excellent mechanical properties is limited to some conventional heat treatment methods. Our previous studies [11,12] showed that the reverse transformation through lamellar phase ($hcp_2$) to austenite resulted in grain refinement by an order of 1/10. The proposed method promoted a microstructure containing a fine equiaxed grain distribution. This new grain refinement process is similar to the reverse transformation in steels [13–15].

The aim of the present study is to induce greater volume fraction of athermal $\varepsilon$ martensite embryos through introducing thermal stresses via cyclic solution heat treatment (repetitive heating and cooling cycles). This new approach for solution heat treatment not only increased the amount of athermal $\varepsilon$ martensite, but also caused a reverse transformation (from martensite to austenite) during the next solution cycle leading to semi-equiaxed secondary austenite grains forming beneath the surface layer. Therefore, this new heat treatment method could be used as a replacement of SPD methods such as surface friction treatment in order to surface grain refinement in Co-Cr-Mo alloy.

## 2. Experimental Procedure

ASTM F75 alloy was purchased from BEGO (Bremen, Germany) and used in the current investigation. The chemical composition of the alloy used in the current study is listed in Table 1.

**Table 1.** Chemical composition of the investigated Co-Cr-Mo alloy (wt.%).

| Cr | Mo | C | Si | Co |
|------|-----|-----|-----|------|
| 28.1 | 5.1 | 0.3 | 1.5 | Rem. |

Solution heat treatment was carried out in a high-temperature tube furnace under an argon (Ar) inert gas atmosphere. For microscopic observation, a conventional metallographic technique was employed. Figure 1 shows the schematic illustrations of the given solution heat treatments used in this investigation. The samples were separated into two groups: (I) the first group of samples was solution-treated at 1220 °C for 3 and 5 Continuous hours (from now on labeled as 3C and 5C) followed by quenching in water, and (II) the second group of samples was solution treated for 3 and 5 discontinuous hours (cyclic treatment, from now on designated by 3S and 5S codes) followed by quenching in water. At the end, all the solution heat-treated samples were isothermally aged at 850 °C for 10 h.

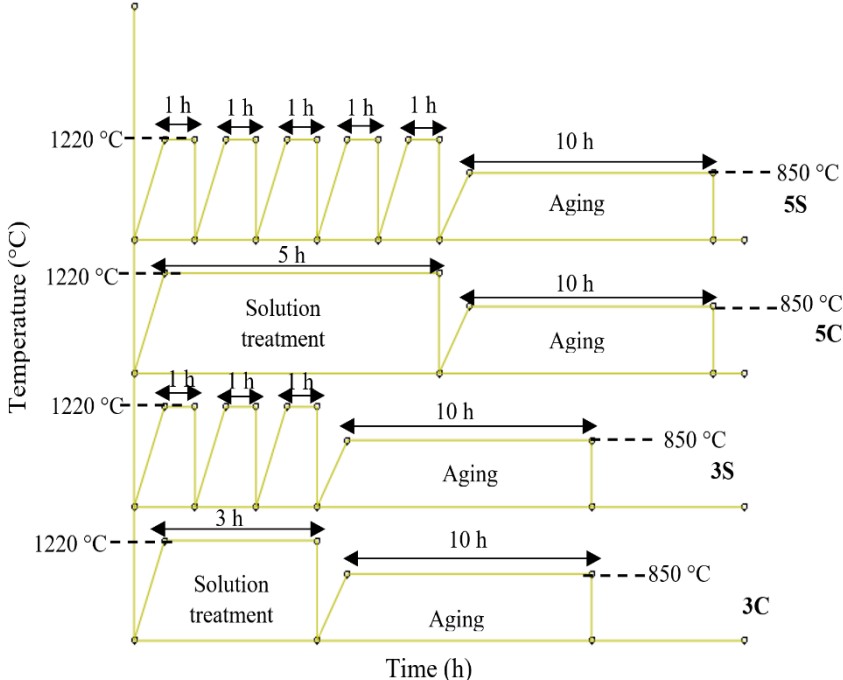

**Figure 1.** Schematic representations of the thermal cycles employed in the current research.

The phase identification was carried out using X-ray diffraction technique with Cu K$\alpha$ ($\lambda$ = 1.54184 Å) from 2$\theta$ = 40–55° with step size of 0.02° 2$\theta \times$ S$^{-1}$. The amount of transformed hcp ($f^{hcp}$) (during quenching and isothermal aging) and untransformed fcc ($f^{fcc}$) were estimated by measuring the integrated intensities of $(200)_{fcc}$ and $(10\bar{1}1)_{hcp}$ peaks. The weight fraction of transformed hcp phase was calculated using the following equation proposed by Sage and Gilluad [16].

$$f^{hcp}_{(wt\ pct)} = \frac{I^{hcp}_{(10\bar{1}1)}}{I^{hcp}_{(10\bar{1}1)} + 1.5I^{fcc}_{(200)}} \tag{1}$$

The focused ion beam milling (FIB) method was employed to make samples for Transmission Electron Microscopy (TEM) studies. All the TEM samples were taken from two regions including inside and near the surface sample. Transmission Electron Microscopy (TEM) (FEI Company, Oregon, USA) was conducted using a Philips CM-200 (FEI Company, Hillsboro, OR, USA) field emission gun operated at 200 kV. For microstructural study, all specimens were mounted in the resin epoxy and polished to mirror-like finish in colloidal solution and cleaned in an ultrasonic bath for 10 min. The etchant used to reveal the microstructure was 92 pct HCl, 5 pct H$_2$SO$_4$ and 3 pct HNO$_3$. Field-emission Scanning Electron Microscopy (NanoSEM 450) (FEI Company, Hillsboro, OR, USA) equipped with Energy Dispersive X-ray Spectroscopy (EDS) (FEI Company, Hillsboro, OR, USA) was used for material characterization. The Potentiodynamic polarization analysis was conducted using an Autolab PGSTAT

302N potentiostat (Metrohm Autolab, Utrecht, Netherlands). Potentiodynamic polarization curves were obtained by sweeping the potential from −1 V to 2 V at a scan rate of 20 mV·s$^{-1}$. The platinum and Saturated Calomel Electrode (SCE) were used as a counter electrode and reference electrode, respectively. All the experiments were performed in 3.5 wt.% NaCl solution with pH equal to 7 (electrolyte) at room temperature and repeated three times to ensure the consistency.

## 3. Results and Discussion

### 3.1. Initial Microstructure (As-Cast Condition)

The initial microstructure of the as-received alloy in the as-cast condition is shown in Figure 2a,b. As can be seen, the initial microstructure of the as-cast alloy contains coarse dendritic structure with interdendritic carbides precipitated during solidification. In general, the strengthening mechanisms in Co-Cr-Mo alloys include: (I) solid-solution strengthening, (II) grain refinement, (III) strain hardening and interaction of dislocations with stacking faults, grain boundaries, twin boundaries, etc., and (IV) precipitation of $M_{23}C_6$ carbides (M = mostly Cr and Mo) acting as reinforcing agents. Figure 2c,d show the SEM micrograph of the precipitated $M_{23}C_6$ carbides present in the alloy matrix at the grain boundary. The EDS point analysis (Figure 2c) and X-ray map elemental analysis (Figure 2d) show that the precipitated carbides at the grain boundary are enriched in Cr and Mo elements. The formation of various carbides with different morphologies and sizes in a cobalt-base alloy depends mainly on the chemical composition, thermal treatment, and cooling rate [17]. The $M_{23}C_6$ and $M_7C_3$ are two quite common carbides that can be found in Co-Cr-Mo (M and C represents metal and carbon atoms, respectively) which results in outstanding wear resistance [18]. As seen in Figure 2d, carbides either precipitated at the grain boundaries or within the grains on dislocations or stacking faults.

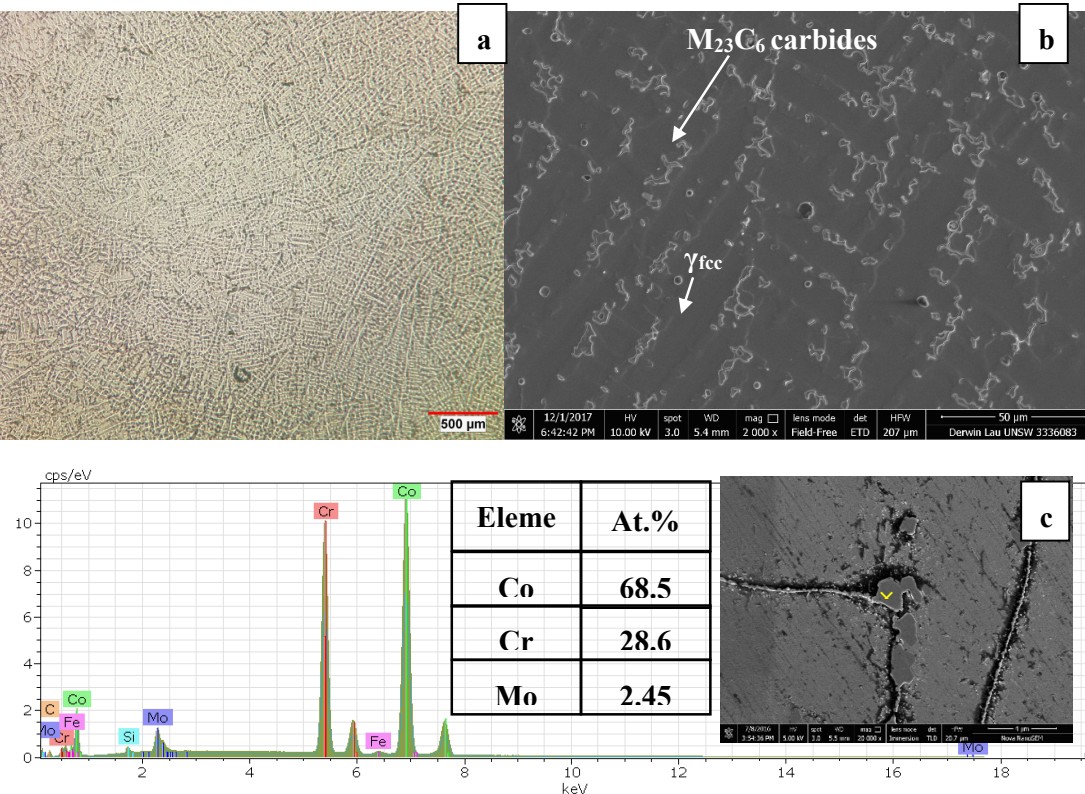

**Figure 2.** (**a**,**b**) initial microstructure of the as-received alloy. (**c**) EDS point analysis of the precipitated carbides, the "⊗" represents the area where the point analysis was taken from. (**d**) X-ray map analysis of carbides.

### 3.2. Solution Heat Treatment

The microstructures of the 3C and 5C samples are shown in Figure 3a,c. As one can see, the solution heat treatment resulted in the dissolution and spheroidization of carbides leading to the homogeneity of the microstructure. The increase of solution time from 3 h to 5 h coarsened the grain size slightly but volume fraction of carbides almost remained unchanged. The increase of solution treatment time from 3 h to 5 h increased the grain size from $\approx$ 210 µm to $\approx$ 520 µm. The presence of parallel lines in the matrix is indicative of athermal $\varepsilon$ martensite (hcp$_1$). According to the optical images, it seems that 5C sample has more athermal $\varepsilon$ martensite than 3C sample. According to Huang's work [19], the volume fraction of athermal $\varepsilon$ martensite was found to be smaller in fine grained alloy, but it concurrently increased with increasing grain size as confirmed in the present work (compare Figure 3a,c). This can be attributed to the increase in the number of probable hcp embryos and lattice defects (twins and stacking faults) with increasing grain size and decrease of M$_s$ temperature with decreasing grain size [19].

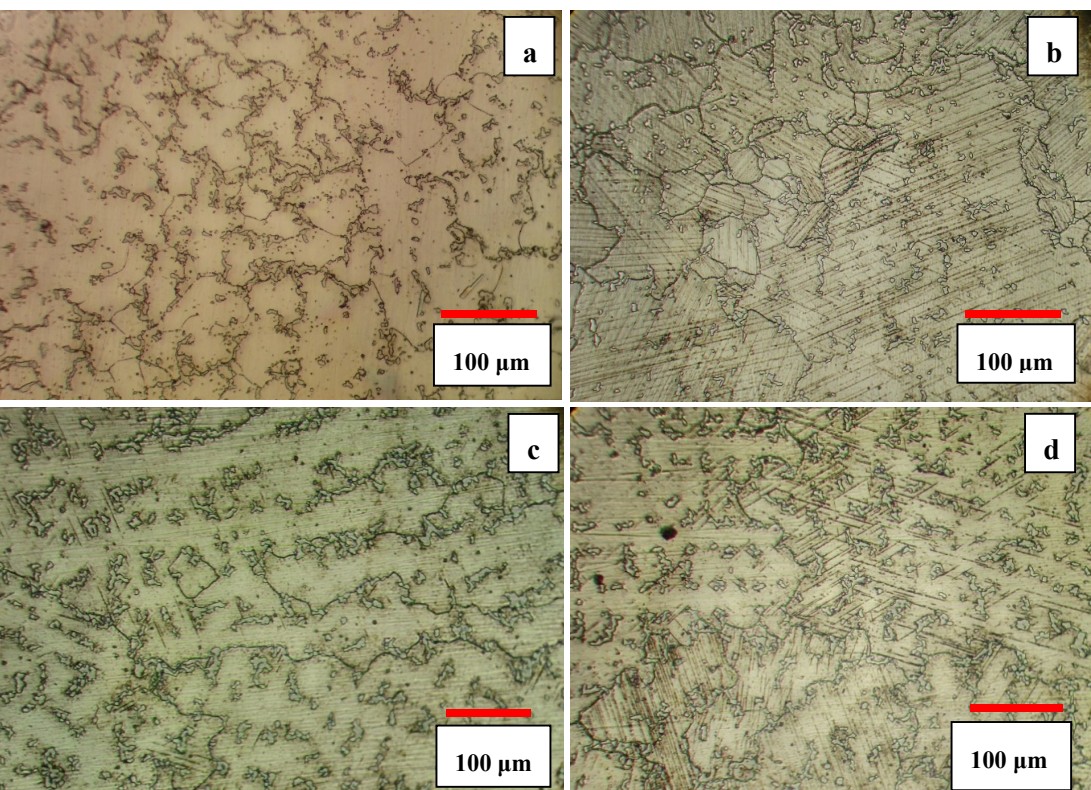

**Figure 3.** Optical micrographs of solution heat-treated samples (**a**) 3C, (**b**) 3S, (**c**) 5C, and (**d**) 5S.

Cyclic heat treatment significantly increased the volume fraction of athermal $\varepsilon$ martensite, as shown in Figure 3b,d, and also undissolved carbides can still be seen within the matrix and at the grain boundaries. It seems that the cyclic heat treatment was more effective in promoting $\gamma_{fcc} \rightarrow \varepsilon_{hcp}$ athermal martensite and provided more nucleation sites for $\varepsilon$ martensite embryos. The thickness of the intergranular $\varepsilon$ martensite varies in different regions and is closely related to the grain size; it was observed that the amount of athermal $\varepsilon$ martensite in 5S sample was greater than that of the 3S sample. The increase of solution time and the number of quenching cycles increased the grain size and provided more nucleation sites for $\varepsilon_{hcp}$ embryos (such as stacking fault defects) inducing more athermal $\varepsilon$ martensite.

The bright field TEM image of the developed microstructure in sample 5S is shown in Figure 4 in which cyclic quenching led to the generation of a series of parallel bands (intergranular striations) within the initial fcc grains. The angle between two athermal $\varepsilon$ martensite bands is around 70° which is

the same as the ideal angle of 70.52° between $(111)_{fcc}$ planes on which $\varepsilon$ martensite forms in accordance with Shoji–Nishiyama orientation relationship [20].

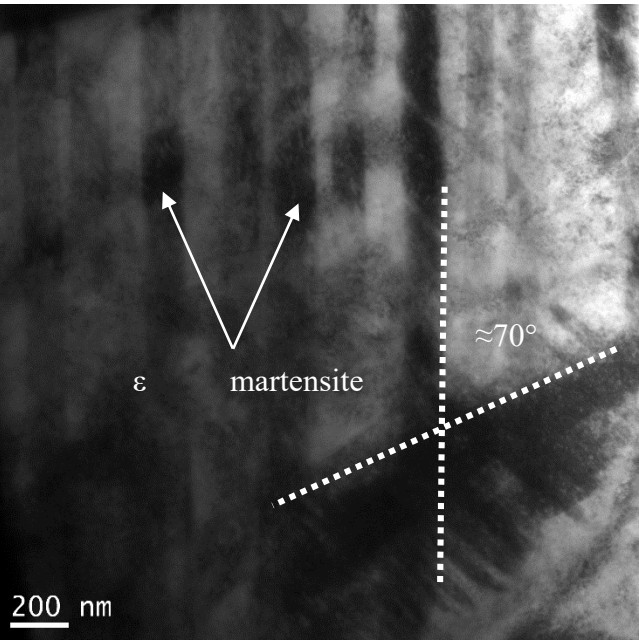

**Figure 4.** TEM bright-field image of primary martensite in the 5S sample showing two athermal martensite with intersection angle of 70°.

Figure 5 Shows the XRD patterns of the solution-treated samples under different conditions. The $(111)_{fcc}$ and $(200)_{fcc}$ diffraction peaks related to the fcc phase appeared at 2θ values of 43.75° and 50.75°, respectively. For the hcp phase, $(10\bar{1}1)_{hcp}$, $(0002)_{hcp}$ and $(10\bar{1}0)_{hcp}$ peaks are located at 2θ values of 41°, 43.75°, and 46.75°, respectively. It was observed that cyclic solution heat treatment in sample 3S resulted in increasing the intensity of $(10\bar{1}1)_{hcp}$ peak and decreasing the intensity of $(200)_{fcc}$ peak. The volume fraction of the hcp phase ($f^{hcp}$) was calculated quantitatively using Equation (1), and it was found that the volume fraction of the athermal $\varepsilon$ martensite increased from 36% in sample 3C to 64% in sample 3S. It is interesting to note that the intensity of $(111)_{fcc}$ peak for this sample nearly remained unchanged. The increase of quenching cycles from 3 to 5 increased the volume fraction of the athermal $\varepsilon$ martensite. As one can see, $(200)_{fcc}$ peak is almost vanished in 5C and 5S samples. In addition, new $(0002)_{hcp}$ peak appeared at 2θ value of 43.75°. Using the same equation, the volume fraction of the athermal $\varepsilon$ martensite in samples 5C and 5S was 81% and 90%, respectively. These results are in good agreement with the microstructural observation which was shown earlier.

Our previous studies [8,9] showed that quenching the Co-Cr-Mo alloy after solution treatment above 1100 °C promoted the formation of athermal $\varepsilon$ martensite. It seems that the formation of high temperature defects which could act as potential sites for the nucleation of $\varepsilon$-martensite embryos is strongly favored at high cooling rates [21]. In addition, the high cooling rate generates internal residual stress resulting in the partial transformation of metastable fcc → hcp $\varepsilon$ martensite.

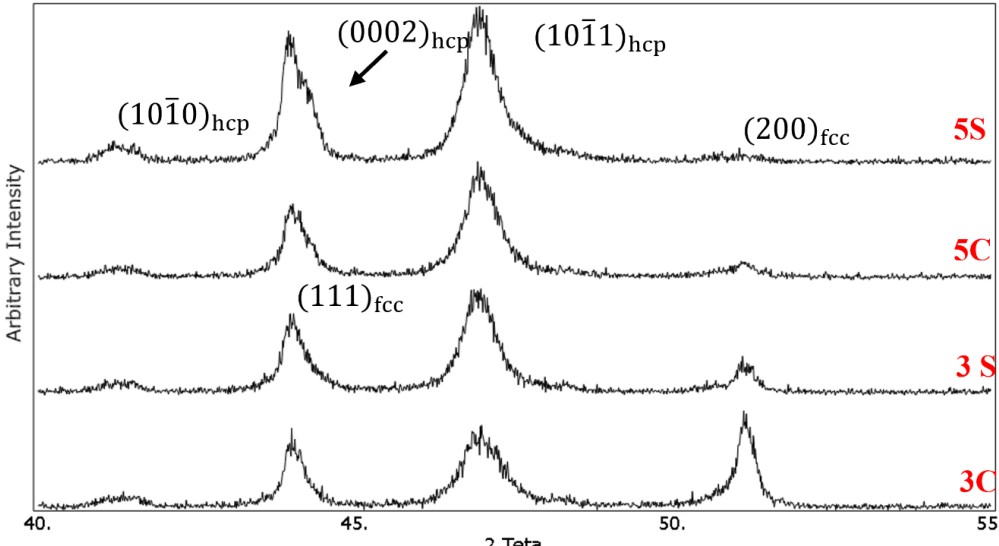

**Figure 5.** X-ray diffraction patterns of the solutioned treated samples, 3C: solutionised continuously for 3hr, 3S: solutionised cyclically for 3 h with an interval of 1hr, 5C: solutionised continuously for 5hr, 5S: solutionised cyclically for 5 h with an interval of 1 h.

### 3.3. Isothermal Aging

In Co-Cr-Mo alloys, martensitic phase transformation can be induced in three different ways: (I) athermal martensite (hcp$_1$), (II) isothermal martensite (hcp$_2$, forming lamellar structure), and (III) stress-induced martensite. The formation of athermal martensite mainly depends upon the solution temperature rather than the holding time at this temperature and in most cases the volume fraction of this phase does not exceed 20% [22]. This was attributed to the lack of enough defects necessary for spontaneous nucleation of ε martensite embryos during quenching [21]. For this reason, isothermal phase transformation is usually conducted at temperatures within the range of 650–950 °C to complete the γ$_{fcc}$ to ε$_{hcp}$ phase transformation. All the solution-treated samples were isothermally aged at 850 °C for 10 h to form isothermal martensite and complete the martensitic phase transformation. The optical images of the isothermally aged samples are shown in Figure 6a–d. No sign of perlite-type lamellae (hcp$_2$ phase) was observed in the matrix after isothermal aging and severe precipitation of M$_{23}$C$_6$ carbides was found to occur on the stacking faults. The kinetic of the isothermal fcc to hcp phase transformation is affected by the amount of lattice defects formed upon quenching. In other words, the increase of athermal martensite as a result of the cyclic solution treatment in the present work has slowed down the formation of perlite-type lamellae in the matrix. This could explain the absence of lamellae phase even after isothermal aging for 10 h at 850 °C. As shown earlier in XRD results, the cyclic solution treatment showed larger volume fraction of athermal martensite as compared to continuously solutionised samples. Massive and dense intergranular striations within the fcc grains are observed in the isothermally aged samples subjected to the cyclic solution treatment.

It is worth noting that the 5S sample shows higher amount of fcc (metastable) → hcp phase than 3S sample. It appears that the increase in the number of cyclic quenching contributed to the formation of heavily faulted regions and intersection of striations acted as potential nucleation sites for the precipitation of fine M$_{23}$C$_6$ carbides (see Figure 6d). The M$_{23}$C$_6$ type carbides (which are fcc type with a lattice parameter similar to the fcc cobalt) are the typical carbides precipitating because of the minimal coherency strains. The intersection of these striations is shown in the bright field TEM image in Figure 7a. It should be noted that these intersections are considered as dislocation dipoles whose high strain fields hinder the motion of slip dislocation [23,24]. The interaction between dislocations and these striations can influence the work hardening behavior of the Co-Cr-Mo alloy leading to the increase of hardness and strength.

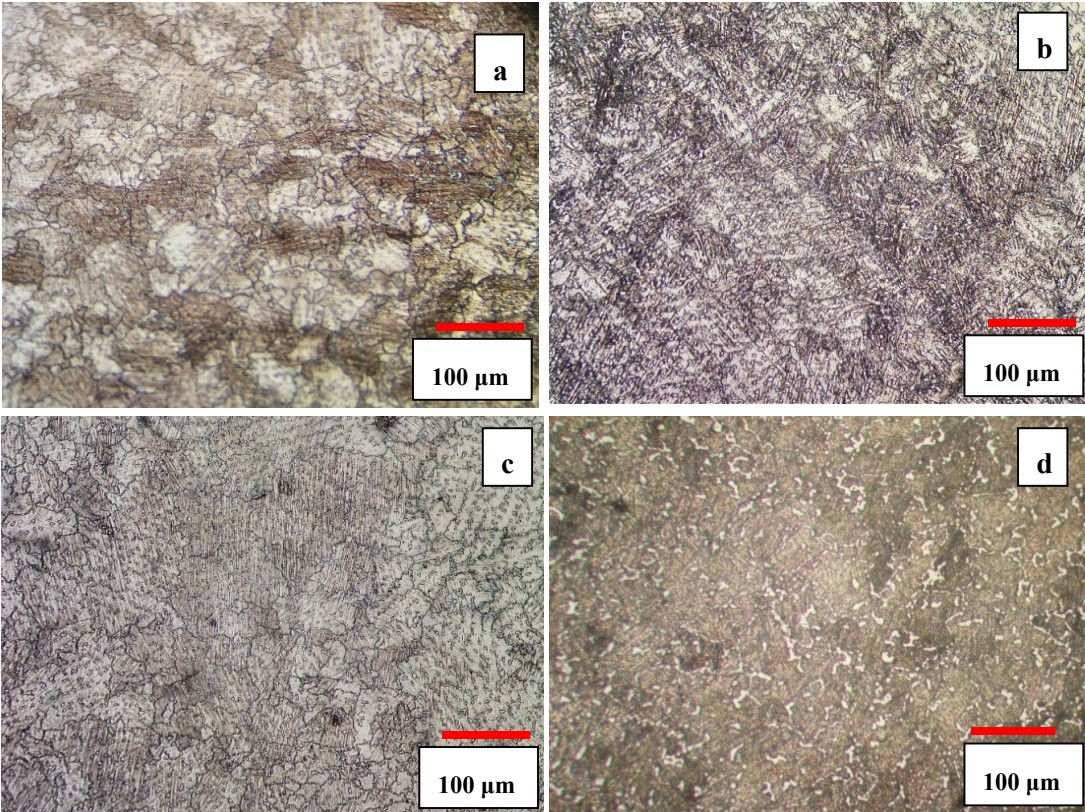

**Figure 6.** Optical micrographs of solution heat-treated samples: (**a**) 3C, (**b**) 3S, (**c**) 5C, (**d**) 5S after aging at 850 °C for 10 h.

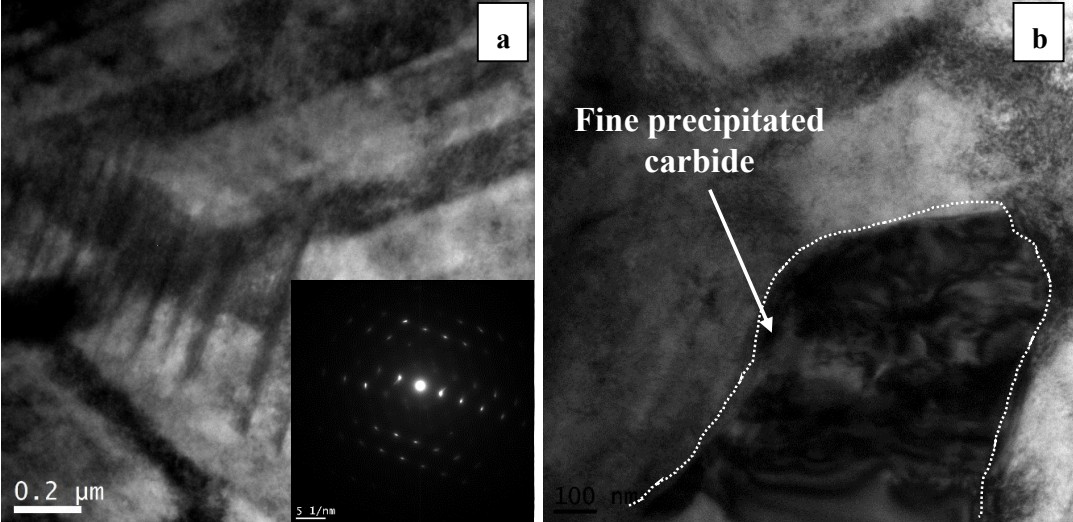

**Figure 7.** TEM bright-field image of the 5S sample (cyclically solutionised at 1220 °C for 5 h with an interval of 1hr followed by isothermal aging): (**a**) formation of heavy faulted regions and striation bands showing ε martensite, and (**b**) discontinuous precipitation of Cr-rich fine carbides.

Figure 7b shows the discontinuous precipitation of fine carbides on the intersection of striations (known as solute drag). This precipitation occurs as a result of preferential diffusion of elements to the faulted regions. Solute atoms such as carbon can migrate to the dislocations and stacking faults leading to the decrease of the stacking fault energy and widening the width of the stacking fault. This microsegregation which is known as Suzuki segregation can effectively cause solid-solution

hardening [25]. Having said that, the diffusion of carbon and chromium into the intersection of striations caused the precipitation of carbides and these carbides interact with the stacking faults locking the dissociated Shockley partial dislocations via solute segregation, as shown in Figure 7b.

The XRD patterns of the isothermally aged samples at 850 °C for 10 h are shown in Figure 8. In both 3C and 5C samples, similar to our previous work [8], isothermal aging led to development of isothermal martensite. This can be seen by the increased intensity of $(10\bar{1}1)_{hcp}$ peak. In addition, after isothermal aging, the $(10\bar{1}1)_{hcp}$ peak in samples 3S and 5S shifted to the right which can be explained by the precipitation of carbides and decrease of lattice d-spacing. This peak shift was not noticeable after solution treatment in Figure 5. The diffusion of Cr from the martensite and precipitation of Cr-rich carbides during isothermal aging decreased the lattice parameter of hcp phase shifting the diffraction peak towards higher angles. The slight peak broadening observed in the cyclically heat-treated samples is attributed to the accumulated residual stress introduced during cyclic quenching. As one can see, the martensitic phase transformation in continuous solutionised samples is not completed, since $(200)_{fcc}$ and $(111)_{fcc}$ peaks did not disappear thoroughly. Applying cyclic heat treatment in samples 3S and 5S resulted in the formation of almost fully martensitic structure as the intensity of $(200)_{fcc}$ and $(111)_{fcc}$ peaks decreased significantly in samples 3S and 5S after 10 h isothermal aging, as shown in Figure 8. According to our previous work [8], isothermal aging at 850 °C for 24 h in the solution-treated samples at 1230 °C for 3Continuous hours, did not result in complete martensite transformation (similar to the results obtained here). It seems that one quenching cycle only led to the formation of slight amount of hcp martensite. However, in this work, applying several cyclic quenching not only increased the volume fraction of athermal martensite but also contributed to the formation of heavily faulted regions providing nucleation sites for isothermal martensite and precipitation of fine carbides on the stacking faults. The effect of this process is alike the formation of stress-induced martensite (plastic straining of 10% and 20% via rolling) in our previous work [9], in which complete martensitic phase transformation was observed just after isothermal aging at 850 °C for 8 h.

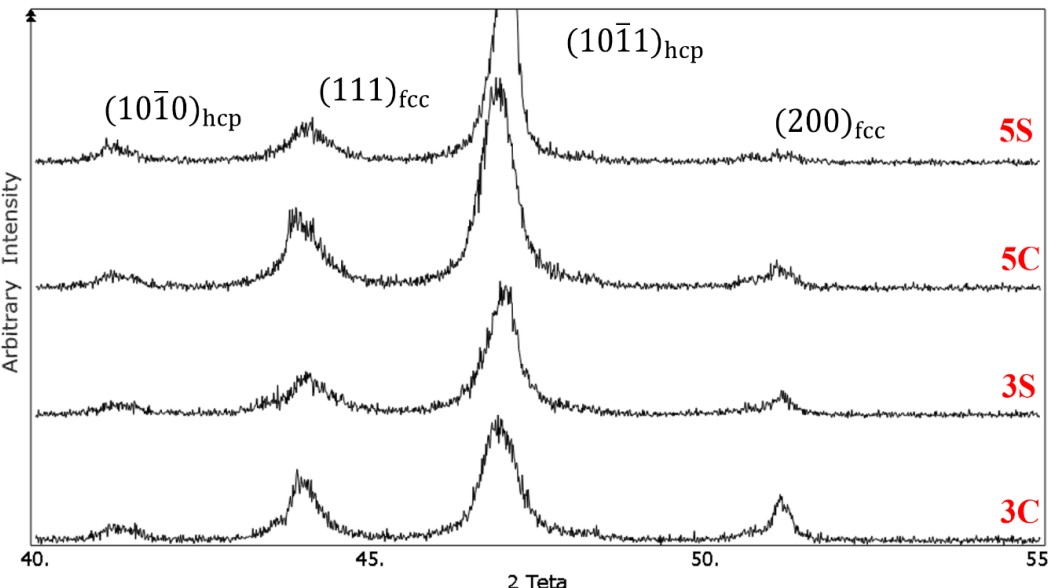

**Figure 8.** X-ray diffraction patterns of the solution-treated samples after aging at 850 °C for 10 h, 3C: solutionised continuously for 3 h, 3S: solutionised cyclically for 3 h with an interval of 1hr, 5C: solutionised continuously for 5 h, 5S: solutionised cyclically for 5 h with an interval of 1 h.

### 3.4. Grain Refinement by Reverse Transformation

Figure 9a–d show the near-surface microstructures of different solution-treated samples after aging at 850 °C for 10 h. In samples 3S and 5S, a new type of grain refinement was observed similar

to what we reported before [11,12]. In our previous works [11,12], austenite grain refinement was achieved through a two-step heat treatment process in which pearlitic-type constitutes ($M_{23}C_6$ + $hcp_2$ martensite, lamellar structure) firstly developed during isothermal aging and then the interface of the lamellar phase was used as the initiation sites for the reverse austenite phase transformation at temperatures 1000–1240 °C. However, in the present work, it seems reasonable to suppose that the nucleation sites for reverse austinites and its concomitant grain refinement method was different than what we observed and reported in our previous studies.

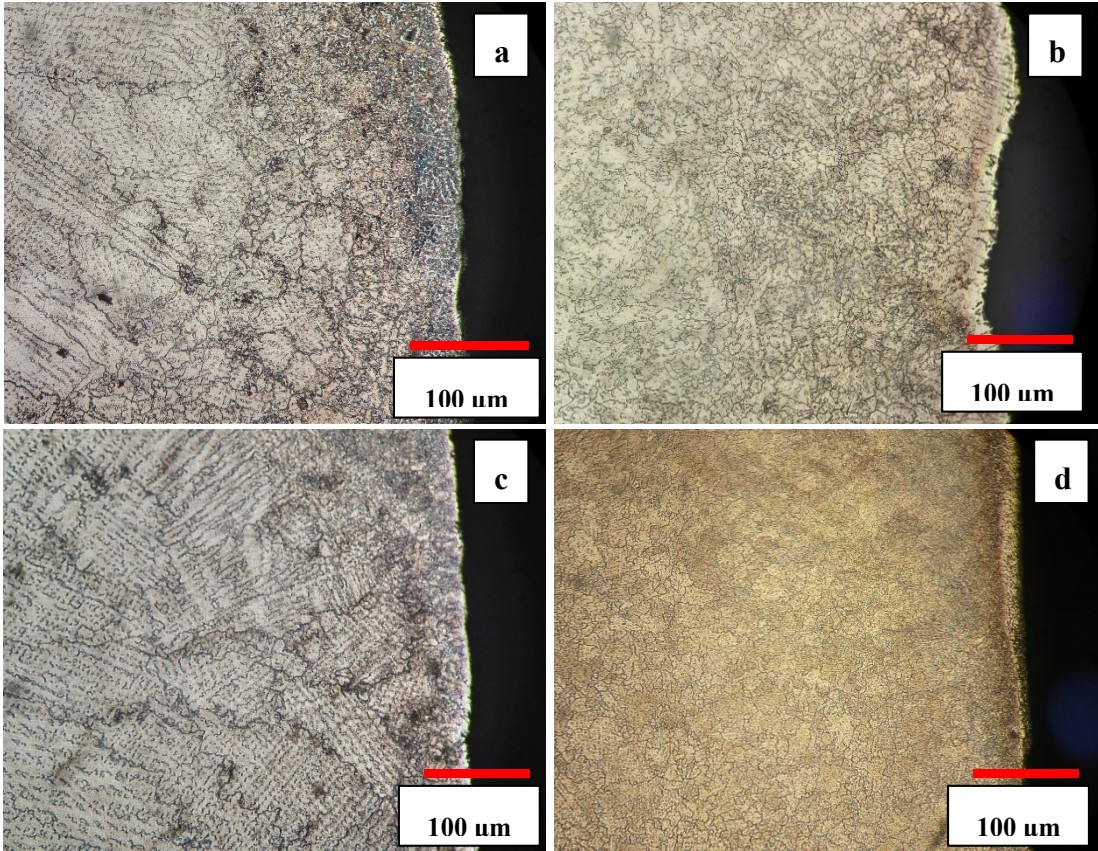

**Figure 9.** Near-surface microstructure of the solution heat-treated samples: (**a**) 3C, (**b**) 3S, (**c**) 5C, (**d**) 5S after aging at 850 °C for 10 h.

In consecutive cyclic quenching, the interface of athermal ε martensite was harnessed to induce a new type of grain refinement and develop new refined austenite grains. Figure 9b and d show 3S and 5S samples aged for 10 h. As one can see, the severity of grain refinement is significant beneath the surface and decreases by moving away from the free surface. The grain size measurement showed that the average grain size decreased from 283 ± 4 μm in as-cast sample to 16 ± 7 μm in 5S sample. The grain refinement is attributed to the generation of the residual thermal stress, mostly near the free surface during cyclic quenching. The increase of number of cyclic quenching resulted in more severe grain refinement and uniform distribution of carbides in the matrix. In addition, it was observed that the depth of the affected area increased by increasing the number of cyclic quenching.

The depth of the affected area in 5S sample was within the range of 400–500 μm, as shown in Figure 10. Cyclic quenching and imposing thermal stresses particularly near the surface have been quite effective and refined the grain size near the surface which can potentially lead to enhanced surface properties such as hardness whereas the inner core material remains unaffected.

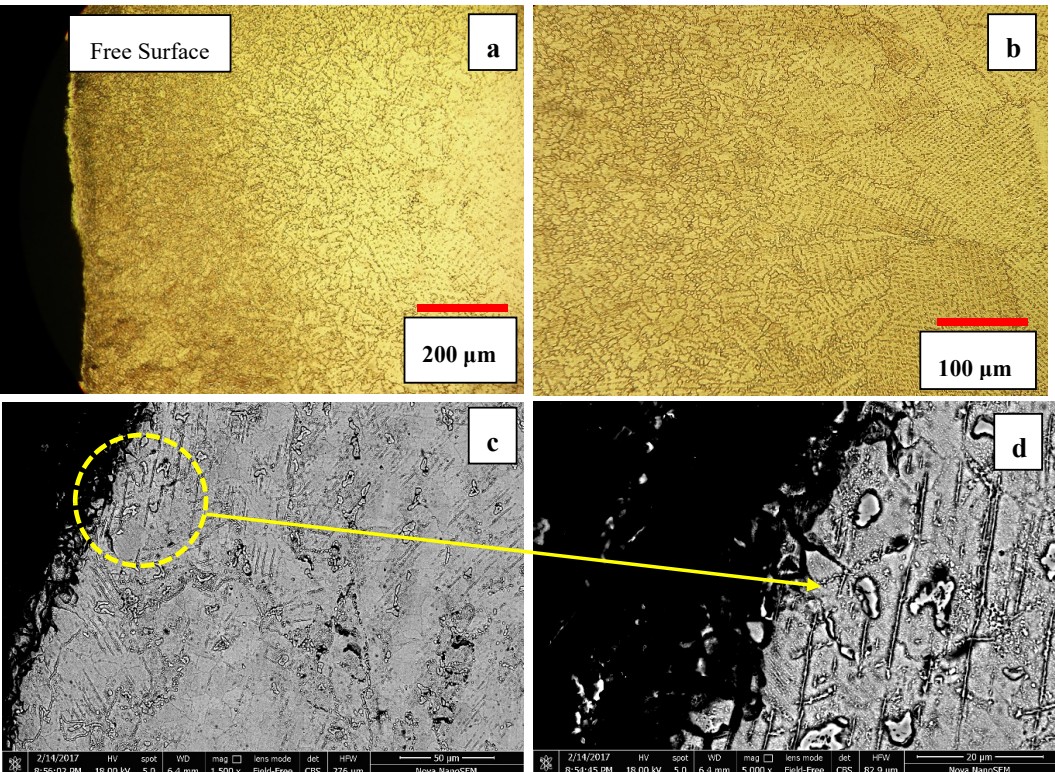

**Figure 10.** Optical micrograph of (**a**) near surface microstructure in 5S sample (**b**) the transition zone shows both fine and coarse grains (**c,d**) SEM micrographs of grain refinement along with martensite development.

One may attribute the observed grain refinement to the recrystallization process. The origin of a recrystallized grain is always a pre-existing area which is highly oriented or heavily faulted in relation to the surrounding material. In metals with a low value of $\gamma_{SFE}$, the difficulty of cross slip and/or climbing of dislocations reduces the ability of the material to accommodate plastic deformation by slip alone (basic mechanisms responsible for recovery), and therefore deformation twinning may occur. The development of subgrains are not usually seen in metals with lower stacking fault energy such as stainless steel and Cobalt-base alloys, because recrystallization usually occurs prior to significant recovery. The other possibility to explain the grain refinement mechanism is the nucleation of the reversed austenite ($\gamma_{fcc}$) on the athermal ε martensite phase which is heavily faulted and can provide the potential nucleation sites for the new fcc grains and lead to the grain refinement near the surface layer, as shown in SEM micrographs in Figure 10c–d. Severe precipitation of $M_{23}C_6$ carbides on the heavily faulted region (ε martensite) and considerable grain refinement of surface layer can lead to the enhanced surface properties such as hardness and wear resistance.

The finite element analysis (FEA) using an Abaqus software was utilized to estimate the amount of thermal stress induced during three cycles of quenching. The sample was initially heated to 1220 °C to make it homogenies and stress-free. The surface was then quenched in cold water until the sample reached the temperature of 5 °C. The current analysis was carried out in Abaqus/Standard through sequential thermal-stress solution procedure. In the sequential thermal-stress simulation, first, transient heat transfer analysis was performed, and then nodal temperatures were used as a loading of thermal-stress analysis. The sample is modeled with three-dimensional continuum elements which consists of 1000 first-order brick elements. Figure 11 shows the effective stress developed after quenching from 1220 °C. It should be noted that for decreasing the computational cost due to the symmetry of the cubic sample, only 1/8 of sample was considered for the simulation. As one can see from Figure 11, stress distribution after quenching is not uniform. The severity of introduced stress is maximum in the middle of the sample edge. It seems that the residual thermal stresses are

accumulative during cyclic quenching. In other words, the thermal stresses build up in the high stress regions and this would provide nucleation sites and enough driving force for the athermal martensitic phase transformation.

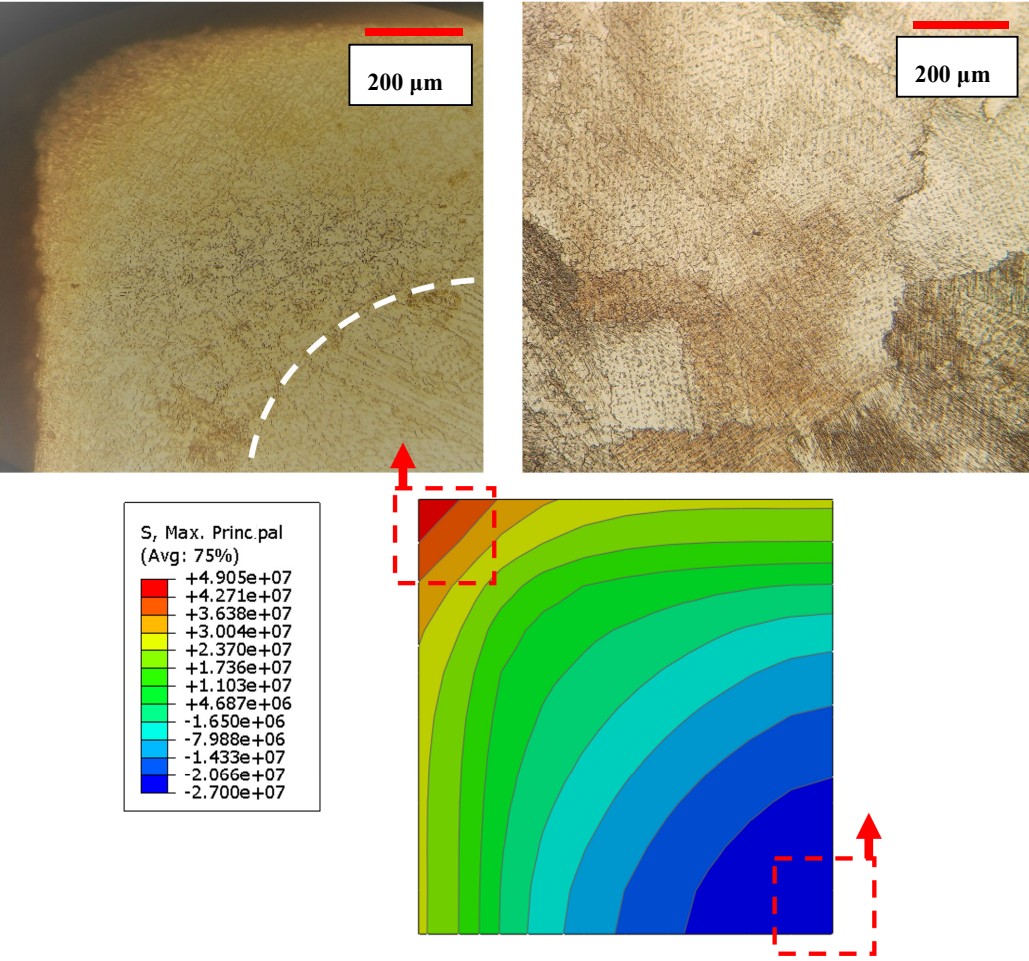

**Figure 11.** Effective stress distribution after 3Cycle quenching, note that grain refinement occurred beneath the surface where severe thermal stress induced.

Figure 12 shows the Tafel plot curves of the samples subjected to different thermal treatment conditions in the 3.5 wt.%. NaCl aqueous solution. Table 2 lists the potentiodynamic polarization data attained from Figure 12 including corrosion current density, polarization resistance, Tafel constants, and corrosion rate. The open circuit potential (OCP, $E_{corro}$) of the as-cast sample is more negative than other thermally treated samples. As one can see, the corrosion resistance of all thermally treated samples is better than the as-cast sample. In other words, the as-cast sample shows higher anodic reactivity compared to other samples. This could be attributed to the coarse dendritic structure with interdendritic carbides precipitated during solidification (see Figure 2). During anodic cycle of the Tafel plot, the passivation behavior was observed in the as-cast sample (from 0.1 to 0.8 V) whereas the current density of all other thermally treated samples was increased gradually before converging and following the same trend. According to the anodic part of the Tafel plot, the passivation behavior began earlier in isothermally treated samples 3C and 5C as compared to other isothermally treated samples (the slope of anodic curves was not as steep as other samples). Stable current density (from 0 to 0.8 V) in the as-cast sample shows the formation of dense protective oxide layer on the surface. At 0.8 V, the current density starts to increase sharply which is related to the passive-transpassive oxidation. The reaction of $Cr_2O_3$ with hydrogen reactivates the anodic dissolution of passive layer leading to oxide layer breakdown ($Cr_2O_3 + 6H^+ \rightarrow 2Cr^{3+} + 3H_2O$) [26,27].

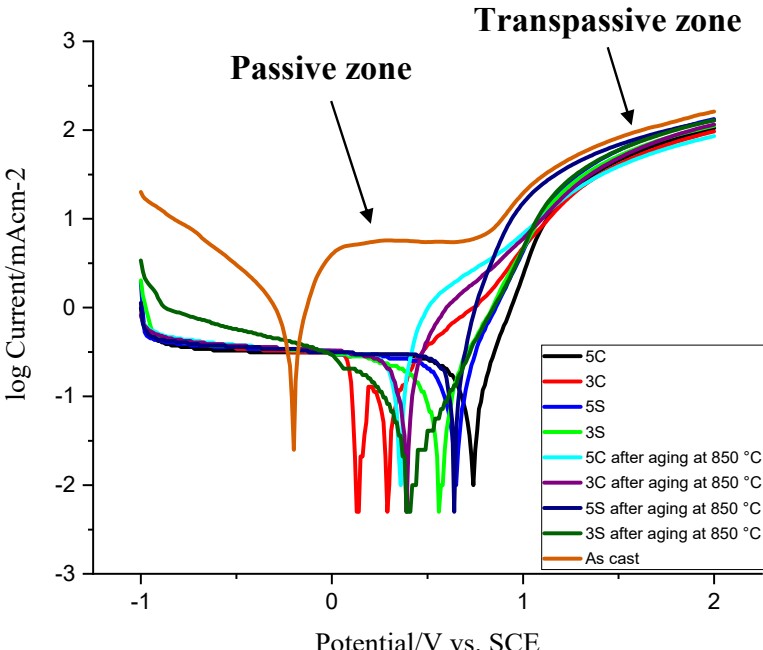

**Figure 12.** Potentiodynamic polarization curves in the 3.5 wt.%. NaCl aqueous solution.

It was shown that the protective oxide layer in Co-Cr-Mo alloy in simulated physiological solution (SPS) at E < 0.3 V consists mainly of $Cr_2O_3$ and the fraction of CoO increased with potential and, in the region from 0.5 to 0.7 V, the fractions of Cr and Co were found to be approximately the same [26]. Generally, all thermally treated samples showed lower corrosion current density and higher polarization resistance as compared to the as-cast sample. This indicates the rate of electrochemical reaction is slow because of the dissolution and spheroidization of carbides, reduction of carbide size, less dendritic microstructure, and homogenization of the microstructure. Among all solutionised samples, sample 3S showed better corrosion behavior ($\approx$1/10 of the as-cast sample). No features or spikes (indicative of the protective oxide layer breakdown) were observed during scan which indicates good corrosion resistance of the thermally treated samples. Isothermal aging at 850 °C for 10 h decreased the corrosion resistance of 5C and 5S samples by $\approx$ 96% and 13%, respectively. However, it improved the corrosion resistance of 3C and 3S samples by $\approx$ 6% and 38%, respectively. It should be noted that the corrosion resistance of all isothermally aged samples is still much better than the as-cast sample which contains coarse dendritic microstructure, microsegregation, and coarse $M_{23}C_6$ carbides in the matrix and on the grain boundaries. As discussed earlier, the amount of athermal $\varepsilon$ martensite is favored by the cyclic solution heat treatment. A maximum volume fraction of 64% was previously reported in the water atomized Co-Cr-Mo powders [26]. The amount of athermal $\varepsilon$ martensite exceeded 80% in chill casting of Co-Cr alloy [26]. Generally, the amount of athermal $\varepsilon$ martensite is influenced by the annealing temperature and higher annealing temperature leads to greater dislocation and fault density [26]. The development of athermal martensite is restricted by the strain fields at the $\gamma_{fcc}/\varepsilon_{hcp}$ interface or local variations in chemical composition that stabilizes the fcc phase (carbon-enriched regions or Cr and Mo-depleted regions). In the present study, we showed that not only the annealing temperature but also the annealing time (3 h and 5 h) and type of heat treatment (continuous versus cyclic) could affect the volume fraction of athermal $\varepsilon$ martensite and this would influence the corrosion properties of Co-Cr-Mo alloy. In fact, repetitive cyclic heat treatment creates more vacancies and stacking faults as compared to the continuous heat treatment providing more nucleation sites for athermal $\varepsilon$ martensite and leading to larger amount of athermal $\varepsilon$ martensite in the matrix.

**Table 2.** Potentiodynamic polarization data of all the samples prepared from Figure 12.

| Parameter | 5C | 3C | 5S | 3S | 5C after Aging at 850 °C | 3C after Aging at 850 °C | 5S after Aging at 850 °C | 3S after Aging at 850 °C | As Cast |
|---|---|---|---|---|---|---|---|---|---|
| $E_{corr}$/V | 0.775 | 0.31 | 0.69 | 0.59 | 0.39 | 0.45 | 0.66 | 0.56 | −0.18 |
| $I_{corr}$/($\mu$A/cm$^{-2}$) | 110 | 216 | 138 | 90 | 216 | 205 | 156 | 56 | 805 |
| $R_p$/Ohm | 529.5 | 483 | 478.4 | 814.9 | 274.2 | 224.8 | 245.2 | 1314 | 91.68 |
| $b_a$/(V/dec) | 0.165 | 0.136 | 0.177 | 0.224 | 0.149 | 0.112 | 0.1 | 0.215 | 0.247 |
| $b_c$/(V/dec) | 0.707 | 1.193 | 1.079 | 0.713 | 1.639 | 1.974 | 0.763 | 0.81 | 0.545 |
| Corrosion Rate/(mm/y) | 0.3597 | 0.7082 | 0.4516 | 0.2969 | 0.7064 | 0.6696 | 0.5118 | 0.184 | 2.634 |

The decrease of the corrosion resistance in 5C and 5S samples after isothermal aging is attributed to the considerable volume fraction of carbides and ε martensite. It has been accepted that the corrosion rate of Co-Cr-Mo alloy increased with the development of HCP phase in the microstructure. It seems that solution treatment for continuous 5 h or 5 repetitive cycles has led to considerable dissolution of primary $M_{23}C_6$ carbides in the matrix and formation of athermal ε martensite. This athermal ε martensite later provides heterogeneous nucleation sites for the precipitation of secondary carbides in the matrix. Carbides themselves are very good corrosion resistant. However, the depletion of Cr adjacent to carbides leads to localized corrosion (microgalvanic corrosion). This is followed by the formation of pits and crevices in the matrix which accelerate the corrosion and release metal ions in the human body. The size, density and distribution of these carbides could have significant influence on the wear and corrosion properties of Co-Cr-Mo alloy. Generally, Co-Cr-Mo alloy consisting of the $\gamma_{fcc}$ is desirable for ductility and the presence of $\varepsilon_{hcp}$ martensite makes this alloy resistant against wear and corrosion [28]. Montero et al. [29] showed that the kinetics of fcc → hcp phase transformation in low carbon Co-Cr-Mo alloy was considerably affected by the introduction of defects (twins and stacking faults) upon quenching after solution treatment. In other words, if fcc phase transforms into hcp phase during quenching, the kinetics of isothermal martensite transformation ($hcp_2$ phase forming lamellar structure) is significantly slowed down. Unlike Monterlo's study, no lamellar phase was observed in the current work even after isothermal aging at 850 °C for 10 h. This could be due to the fact that cyclic heat treatment has resulted in the formation of considerable volume fraction of athermal ε martensite and this has reduced the kinetics of isothermal martensite.

Upon aging, athermal ε martensite continues to grow and fine carbides begin to precipitate on the ε martensite. In Monterlo's study [29], the corrosion resistance of isothermally aged low carbon Co-Cr-Mo alloy was shown to decrease only in those samples subjected to short period of aging where there were significant volume fractions of lattice defects (hcp embryos). Upon the reduction of lattice defects as a result of phase transformation, corrosion resistance was no longer affected by the fcc to hcp phase transformation since the chemical composition did not change during phase transformation.

The improvement of corrosion resistance in the 3S sample in the present study could be explained based upon the volume fraction of lattice defects and stacking faults formed during cyclic quenching and consumption of these defects due to the precipitation of carbides during isothermal aging and absence of lamellar phase. However, in 5S sample, it seems that applying cyclic solution treatment for five times created more lattice defects and stacking faults compared to sample 3S, and isothermal aging at 850 °C for 10 h was not enough to decrease the density of these defects. This perhaps has led to the deterioration of the corrosion resistance in sample 5S. In addition, the grain refinement, and fine and uniform distribution of the secondary carbides due to the cyclic heat treatment have modified the microstructure of the surface and contributed to the superior corrosion resistance of the cyclic heat-treated samples as compared to the as-cast one. Therefore, cyclic solution treatment is proposed as an effective method to enhance the surface characteristics and biocompatibility of the Co-Cr-Mo implant alloy.

## 4. Conclusions

The results of the present investigation show that cyclic solution treatment leads to considerable grain refinement and this effect is more favorable towards the free surface. Discontinuous solution heat treatment for five cycles creates a considerable volume fraction of lattice defects and stacking faults providing heterogeneous nucleation sites for athermal ε martensite. The severity of grain refinement was much higher beneath the surface of alloy due to thermal stress induced during cyclic quenching (≈49 MPa for three cyclic quenching). The new grain refinement during cyclic quenching is attributed to the nucleation of reverse austenite on the athermal martensite. In addition, the results show that athermal martensite development during cyclic quenching contributes to higher amounts of isothermal martensite during aging treatment at 850 °C for 10 h. This novel heat treatment in Co-Cr-Mo alloy

results in a considerable improvement in corrosion resistance of the thermally treated samples as compared to the as-cast one.

**Author Contributions:** Conceptualization, S.Z. and H.R.L.; methodology, S.Z.; software, S.Z.; validation, S.Z., H.R.L. and S.A.; formal analysis, S.Z. And M.S.; investigation, S.Z., S.A. and S.M.-M.; resources, S.Z.; writing—review and editing, H.R.L.; visualization, S.Z.; supervision, S.Z.; project administration, S.Z.; funding acquisition, S.A. All authors have read and agreed to the published version of the manuscript.

**Funding:** This research received no external funding.

**Acknowledgments:** This research did not receive any specific grant from funding agencies in the public, commercial, or not-for-profit sectors.

**Conflicts of Interest:** The authors declare no conflict of interest.

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
