# Peer review of "The Effect of Cyclic Solution Heat Treatment on the Martensitic Phase Transformation and Grain Refinement of Co-Cr-Mo Dental Alloy"

_metals, doi:10.3390/met10070861_

Round 1
Reviewer 1 Report
At first - the description on part of figures are not enough visible therefore readers can't recognise what authors showed on them e.g. fig. 1 The schematic representations od thermal cycles employed in the current research. is totally not clear and therefore it is hard to discuss the next parts of the manuscript.
Nevertheless, I can suggest just now improving some elements:
1. point 2 Experimental Procedure - there is a lack information about the procedure of preparing samples to SEM and LM observations.
2. fig.2b The description is not enough clear - can authors insert the picture with a little bit bigger magnification?
3. Are authors sure that in the structures we can observe only one type of carbides M23C6?
4. Fig.2. The map of the distribution of particular elements contains only Co, Cr and Si. Why authors analysed Si but not Mo? Do authors have looked for the SiC? The map of Mo has to be inserted.
4.1. Why authors present only EDS spectrum why not the quantity analysis (or semi-quantity) of particular elements?
4.2. EDS analysis of matrix has to be shown to. In the maps have to be attached SEM picture.
4.3. Fig. 2d what kind of SEM detectors was used SE or BSE? It worth to note: to recognise differences in the chemical composition will be better to use BSE.
4.4. The description of Fig.2d "carbides forming on stacking faults" - Are authors sure - taking into account the SEM pictures it can be the only presumption; What kind of carbides can forming in such areas?
5. The fig. 4 is not enough clear. Can authors insert into manuscript electron diffraction picture to confirm the description?
6. Description of picture 5 has to be improved
7. line 201 - "high-resolution TEM image" - fig.7a- unfortunately it is not high-resolution image
8. Fig.7b.- descriptions has to be improve
9. The discussion in point 3.4 have to improve because contain mistakes and misunderstandings sentences e.g.:
9.1 line 261 " ...as materials with high stacking fault energy (such as Al-, Ni-alloys and austenitic stainless steels) - I definitely not agree with authors (!) Aluminium and Al-alloys have high energy of stacking fault approx 240mJ/m2, austenitic steel like 1.4404 have got 10-20 mJ/m2 have low energy of stacking fault.
9.2. Line 261-264 " The reason for this is due to the difficulty of cross slip and/or climbing of dislocations in low stacking fault energy alloys. Therefore, they need a high degree of plastic deformation and annealing at high temperature to be statically or dynamically recrystallized" - can authors discuss this point of view. In my opinion, the statements contain opposite sentence.
9.3. Line 267 Fig.10c,d, unfortunately, it is hard to see the "grain refinement" - can authors changes the pictures into more accurate
10. Taking into account the main idea of considerable grain refinement it needs to measure the size of grains in the initial condition and after proposed treatments.
Author Response
Reviewer 1
At first - the description on part of figures are not enough visible therefore readers cannot recognise what authors showed on them e.g. fig. 1 The schematic representations of thermal cycles employed in the current research. is totally not clear and therefore it is hard to discuss the next parts of the manuscript.
- We have changed Figure 1 to make it clearer, as requested.
Nevertheless, I can suggest just now improving some elements:
Q1. point 2 Experimental Procedure - there is a lack information about the procedure of preparing samples to SEM and LM observations.
A1. Thanks for your comments. We have added a description which explains the steps taken to prepare the samples for microstructural observation, as follows:
“For microstructural study, all specimens were mounted in the resin epoxy and polished to mirror-like finish in colloidal solution and cleaned in an ultrasonic bath for 10min. The etchant used to reveal the microstructure was 92 pct HCl, 5 pct H2SO4 and 3 pct HNO3.
Q2. fig.2b The description is not enough clear - can authors insert the picture with a little bit bigger magnification?
A2. Figure 2 is a combination of optical image and high magnification SEM images with point analysis and X-ray map. High magnification SEM image has already been provided in Figure 2c and 2d, showing the distribution of carbides in the matrix and the fact that they are enriched in Cr (perhaps M23C6 and/or M7C3).
- Are authors sure that in the structures we can observe only one type of carbides M23C6?
A3. M23C6 and M7C3 are two most common carbides in CoCrMo implant alloys. M7C3 carbides usually decompose to M23C6 carbides during solidification. However, M7C3 carbides could also locate within the grains and/or at grain boundaries. Subsequent heat treatments (intentional or from service exposure) modify the morphology, amounts, and types of carbides found in the grains of superalloys. Some secondary carbides can be formed within grains by precipitation on dislocations located near large primary carbides.
Q4-1. Fig.2. The map of the distribution of particular elements contains only Co, Cr and Si. Why authors analysed Si but not Mo? Do authors have looked for the SiC? The map of Mo has to be inserted.
A4-1. Silicon is generally added to increase the fluidity during casting and sometimes it appears as inclusion such as SiO2, as shown in X-ray map. SiC does not form since the affinity of Cr and Mo to form carbides is greater than Si. Since it was difficult to find the exact spot, we replaced the X-ray image with another one.
Q4-2. Why authors present only EDS spectrum why not the quantity analysis (or semi-quantity) of particular elements?
A4-2. The quantity analysis was also added to EDS spectrum.
Q4.3. EDS analysis of matrix has to be shown to. In the maps have to be attached SEM picture.
Q4-3. I believe it is not really necessary to put the EDS of matrix here and increase the number of figures unnecessarily. The X-ray map of the matrix is clearly presented in Figure 2d which shows the presence of Co and Cr in the matrix. This phase is γfcc austenite.
Q4.4. Fig. 2d what kind of SEM detectors was used SE or BSE? It worth to note: to recognise differences in the chemical composition will be better to use BSE.
A4.4. Here is the description of the SEM used in the present study: “The NanoSEM 450 is a field-emission scanning electron microscope (FE-SEM), which attains ultra-high imaging resolution without the specimen size restrictions of a conventional in-lens FE-SEM due to the advanced design of the electron optics. The NanoSEM 450’s Schottky field-emission source allows the user to achieve high imaging resolution at a range of kV, at both low (high-resolution imaging) and high (microanalytical imaging) currents. Secondary electron (SE) imaging can be undertaken in both field-free and immersion mode for comprehensive low-to-high resolution imaging of a variety of samples. The NanoSEM 450 is fitted with a retractable annular backscattered electron detector as well as a Bruker SDD-EDS detector for the convenient visualisation of compositional differences across the specimen surface”.
We have to retract the BSE detector to do EDS analysis otherwise X-rays are blocked by the BSE detector.
Q4.5. The description of Fig.2d "carbides forming on stacking faults" - Are authors sure - taking into account the SEM pictures it can be the only presumption; What kind of carbides can form in such areas?
A4.5. We have changed this Figure, as explained in A4-1. To answer the question:
M23C6 and/or M7C3 secondary carbides can also form on the stacking fault defects and dislocations since these defects provide fast diffusion path for Cr and C. As an example, in our previous studies entitled “Nanoscale carbide precipitation in Co–28Cr–5Mo–0.3C implant alloy during martensite transformation”, we showed that the simultaneous existence of carbon and chromium at the hcp embryos–fcc matrix interfaces and dragging force to attract carbon to dislocation networks contribute to the formation of nanoscale M23C6 carbides at these regions. This has been heavily studied in our previous publications, as listed below:
- Determination of residual stress on TIG-treated surface via nanoindentation technique in Co-Cr-Mo-C alloy
Surface and Coatings Technology, Volume 38025 December 2019Article 125020.
- The effect of friction stir processing (FSP) on the microstructure, nanomechanical and corrosion properties of low carbon CoCr28Mo5 alloy, Surface and Coatings Technology, Volume 35425 November 2018Pages 390-404
- Microstructural characterization of TIG surface treating in Co-Cr-Mo-C alloy, Materials Characterization, Volume 132October 2017Pages 223-229
- Microstructure and tribological characteristics of aged Co–28Cr–5Mo–0.3C alloy, Materials & Design, Volume 37May 2012Pages 292-303
- Effect of isothermal aging on the microstructural evolution of Co–Cr–Mo–C alloy, Materials Science and Engineering: A, Volume 527, Issues 24–2525 September 2010Pages 6494-6500
- Microstructural evolution during isothermal aging and strain-induced transformation followed by isothermal aging in Co-Cr-Mo-C alloy: A comparative study, Materials Science and Engineering: A, Volume 527, Issues 16–1725 June 2010Pages 4082-4091
- Nanoscale carbide precipitation in Co–28Cr–5Mo–0.3C implant alloy during martensite transformation, Materials Letters, Volume 1161 February 2014Pages 188-190
Q5. The fig. 4 is not enough clear. Can authors insert into manuscript electron diffraction picture to confirm the description?
A5. Thanks for your comment. This figure is a high-resolution TEM (HRTEM) image which clearly shows the formation of a series of striations bands indicative of ε martensite. The complexity of carbides makes EBSD a challenging task and that’s why we chose HRTEM.
Q6. Description of picture 5 has to be improved
A6. Thanks for your comment. It has been changed.
Q7. line 201 - "high-resolution TEM image" - fig.7a- unfortunately, it is not high-resolution image
A7. We have changed it to “ TEM bright field” image.
Q8. Fig.7b.- descriptions has to be improved.
A8. Thanks for your comment. It has been changed.
Q9. The discussion in point 3.4 have to improve because contain mistakes and misunderstandings sentences e.g.:
Q9.1 line 261 " ...as materials with high stacking fault energy (such as Al-, Ni-alloys and austenitic stainless steels) - I definitely not agree with authors (!) Aluminium and Al-alloys have high energy of stacking fault approx 240mJ/m2, austenitic steel like 1.4404 have got 10-20 mJ/m2 have low energy of stacking fault.
A9.1. Thanks for your comment. It has been corrected.
Q9.2. Line 261-264 " The reason for this is due to the difficulty of cross slip and/or climbing of dislocations in low stacking fault energy alloys. Therefore, they need a high degree of plastic deformation and annealing at high temperature to be statically or dynamically recrystallized" – can authors discuss this point of view. In my opinion, the statements contain opposite sentence.
A9.2. I believe the statement is true. Here are the reasons:
In Cobalt alloy metallurgy, alloying elements such as iron, manganese, nickel, and carbon tend to stabilize the fcc structure and increase stacking-fault energy, whereas elements such as chromium, molybdenum, tungsten, and silicon tend to stabilize the hcp structure and decrease stacking-fault energy. The solid-solution alloying decreases stacking-fault energy, thereby making the cross slip and climb of glide dislocations more difficult. The cross slip and glide of dislocations being necessary for polygonization is difficult to happen in low-stacking fault energy alloys such as CoCrMo. We have removed the following sentence to avoid confusion:
“Therefore, they need a high degree of plastic deformation and annealing at high temperature to be statically or dynamically recrystallized”. But, in general, we intended to say that high storage energy is needed for recrystallization in low-stacking fault energy alloys. And high storage energy is attained by plastic deformation and cold-working.
9.3. Line 267 Fig.10c,d, unfortunately, it is hard to see the "grain refinement" - can authors changes
the pictures into more accurate
A9. The grain refinement is quite visible in Figure 10b. Figures 10c and d are high-magnification SEM images showing the precipitation of carbides on the stacking fault defects introduced during cyclic quenching.
Q10. Taking into account the main idea of considerable grain refinement it needs to measure the size of grains in the initial condition and after proposed treatments.
A10. The grain size was measured and added into the manuscript. The grain size was reduced from 283±4μm in as-cast sample to 16±7μm in 5S sample.

Reviewer 2 Report
The description in Figure 1, 5, 7, 8, 9, 11 and 12 are illegible.
Author Response
Reviewer 2
Q1. The description in Figure 1, 5, 7, 8, 9, 11 and 12 are illegible.
A1. Thanks for your comment. It has been corrected.
Reviewer 3 Report
This paper reported that the cyclic solution treatment was effective to decrease grain size and to increase the volume fraction of epsilon martensite only near the surface for a Co-28Cr-6M0o-0.3C implant alloy. The quality of micrographs is poor and hence it is difficult to follow the explanation. Additional experimental data are also needed to claim the usefulness of cyclic solution treatment. Discussing points include,
- First of all, please show the depth profile of carbon concentration in the heated samples. The decarburization during cyclic heating and aging would be suspected to lower the stability of austenite near the specimen surface.
- Some labels are missing in several figures. Were there any problems to convert original drawings to pdf ?
- OM and TEM micrographs are not clear. Please provide EBSD maps to catch microstructural features, particularly in order to discuss the change in grain size?
- Concerning TEM results, please present the diffraction patterns corresponding to epsilon martensite and relevant dark field image to identify epsilon plates. It is also difficult to find the presence of fine precipitate in Fig. 7(b).
- In X-ray diffraction patterns presented in Figs. 5 and 8, the changes in peak positions (peak shifts) and shapes (broadening: FWHM) seem to be observed. These changes should be explained in terms of crystal defects and residual phase or/and intergranular stresses.

Author Response
Reviewer 3
This paper reported that the cyclic solution treatment was effective to decrease grain size and to increase the volume fraction of epsilon martensite only near the surface for a Co-28Cr-6M0o-0.3C implant alloy. The quality of micrographs is poor and hence it is difficult to follow the explanation. Additional experimental data are also needed to claim the usefulness of cyclic solution treatment. Discussing points include:
Q1. First of all, please show the depth profile of carbon concentration in the heated samples. The decarburization during cyclic heating and aging would be suspected to lower the stability of austenite near the specimen surface.
A1. The heat treatment was carried out under an argon protective atmosphere to minimise the oxidation issue. Bear in mind, the formation of chromium oxide on the surface of CoCrMo alloy tends to minimise the decarburization problem. This can be seen in Figure 10 where precipitation of carbides took place just underneath the surface. The main important aspect of the present study is introduction of stacking fault defects via cyclic heat treatment and its concomitant thermal stresses providing nucleation sites for reverse austenite.
Q2. Some labels are missing in several figures. Were there any problems to convert original drawings to pdf ?
A2. Apparently, there was an issue with conversion. We have raised it to the Editor.
Q3. OM and TEM micrographs are not clear. Please provide EBSD maps to catch microstructural features, particularly in order to discuss the change in grain size?
A3. OM images clearly show the grain refinement achieved in the region close to the surface and TEM images show the formation of martensite during cyclic treatment. The grain size was measured and included in the manuscript.
Q4. Concerning TEM results, please present the diffraction patterns corresponding to epsilon martensite and relevant dark field image to identify epsilon plates. It is also difficult to find the presence of fine precipitate in Fig. 7(b).
A4.The precipitate in Figure 7b is marked for clarification. The SAD pattern has been added.
Q5. In X-ray diffraction patterns presented in Figs. 5 and 8, the changes in peak positions (peak shifts) and shapes (broadening: FWHM) seem to be observed. These changes should be explained in terms of crystal defects and residual phase or/and intergranular stresses.
A5. We have added the following description:
In addition, after isothermal aging, the (101 Ì…1)hcp peak in samples 3S and 5S shifted to the right which can be explained by the precipitation of carbides and decrease of lattice d-spacing. This peak shift was not noticeable after solution treatment in Figure 5. The diffusion of Cr from the martensite and precipitation of Cr-rich carbides during isothermal aging decreased the lattice parameter of hcp phase shifting the diffraction peak towards higher angles. The slight peak broadening observed in the cyclically heat-treated samples is attributed to the accumulated residual stress introduced during cyclic quenching.

Round 2
Reviewer 1 Report
Thank you very much for the changes in the manuscript and the "Explanations letter". Unfortunately, the current version of the manuscript has still not enough quality. Please find below points that ought to be corrected:
1. Unfortunately, the authors have still problems with editing mistakes. In the common situation authors before sending the to publisher check carefully the quality of the manuscript(!) e.g.:
- Fig. 2. tab with chemical composition - some data is not enough visible,
- Fig.3. what are the values under markers - I supposed 100nm?
- Fig. 6 what are the values under markers - I supposed 100nm?
- Fig7. In my previous comments - point 7b - authors insert line (I think it is unnecessary) but below word "Fine" the word is not visible - I suppose precipitated (?)
- Fig. 11. What are the values under markers - I supposed 100nm?
- Fig.12. Sentence: "passive zone" is not enough visible.
2. According to comments Q4-1 - The text always should be connected with pictures therefore when authors show Si map should comment it (even shortly). Of course, presence/or not of Si map in the manuscript is the decision of authors.
3. according to comments Q 4.2 - Thank you very much for the insert into the manuscript of EDS quantitative analysis.
I need to short comments on the EDS analysis: When we look for the phases we have to examine all chemical elements which formed them. Therefore I have a question: why the authors didn't examine carbon? If the maps of carbon haven't shown significant changes - the authors should comment on it.
4. According to Q4.5. If we don't have any real proof we can only suppose the existence of them or if the authors have provided examination with the same materials or similar (e.g. in other papers) then authors should insert into text the appropriate reference (!).
5. According to A5 - authors use in the answer the sentence "HRTEM" (i.e. high-resolution image). In the common use, the mentioned HRTEM uses when we can recognise much more details of structure e.g. atomic planes.
6. According to 9.2.
It worth to note the recrystallisation process definitely needs some energy which we introduce into the material by plastic working. The recrystallisation the process which depends on a lot of factors. e.g. according to Boczwar relation, in the pure metals, the recrystallisation processes will start more quickly than alloys counterparts.
The energy of stacking fault of material is smaller the more the stacking fault occurs in it. As a consequence, changing the slip plane (cross slip) is more difficult and, as a result, the material strengthens more.
Therefore sentence "high storage energy is needed for recrystallization in low-stacking fault energy alloys" is very discussion. The present sentence suggests that high -stacking fault energy alloys need low storage energy - but .... isn't true.
The materials with high-stacking fault strengthening much less than low-stacking fault energy materials.
7. According to A9 Thank you very much for the explanation. Unfortunately the sentences from the answer:"...images showing the precipitation of carbides on the stacking fault defects introduced during cyclic quenching" don't agree with the text in the manuscript:
(line 263-267): "The other possibility to explain the grain refinement mechanism is the nucleation of the reversed austenite (γfcc) on the athermal ε martensite phase which is heavily faulted and can provide the potential nucleation sites for the new fcc grains and lead to the grain refinement near the surface layer, as shown in SEM micrographs in Figs. 10 c-d" Authors have to change it.
Author Response
Reviewer 1
Thank you very much for the changes in the manuscript and the "Explanations letter". Unfortunately, the current version of the manuscript has still not enough quality. Please find below points that ought to be corrected:
Q1. Unfortunately, the authors have still problems with editing mistakes. In the common situation authors before sending to publisher carefully check the quality of the manuscript(!) e.g.:
- Fig. 2. tab with chemical composition - some data is not enough visible,
- Fig.3. what are the values under markers - I supposed 100nm?
- Fig. 6 what are the values under markers - I supposed 100nm?
- Fig7. In my previous comments - point 7b - authors insert line (I think it is unnecessary) but below
word "Fine" the word is not visible - I suppose precipitated (?)
- Fig. 11. What are the values under markers - I supposed 100nm?
- Fig.12. Sentence: "passive zone" is not enough visible.
A1. The font size of the table in Figure 2 has been increased. The scale bars in Figures 3 and 6 show 100 microns (The font size has been increased). In Figure 7b, the text is “fine precipitated carbide”. The fond size has been increased. For Figure 11, the scale bar is 200 microns. The font size has been increased to make it clear.
Perhaps there might be an issue when converting to pdf file. We will check twice before sending the file out.
Q2. According to comments Q4-1 - The text always should be connected with pictures therefore when authors show Si map should comment it (even shortly). Of course, presence/or not of Si map in the manuscript is the decision of authors.
A2. Thanks for your comment. We have removed that image and replaced it with another one as shown in Figure 2.
Q3. According to comments Q 4.2 - Thank you very much for the inserting the EDS quantitative analysis. I need to short comments on the EDS analysis: When we look for the phases, we have to examine all chemical elements which formed them. Therefore, I have a question: why the authors did not examine carbon? If the maps of carbon have not shown significant changes - the authors should comment on it.
A3. The X-ray map of Carbon has been added.
Q4. According to Q4.5. If we don't have any real proof, we can only suppose the existence of them or if the authors have provided examination with the same materials or similar (e.g. in other papers) then authors should insert into text the appropriate reference (!).
A4. The precipitation of carbides on defects such as dislocations and stacking faults has been very well studied and cited in the literature including our previous works on CoCrMo alloys. CoCrMo alloy is known to have extremely low stacking fault energy (nearly 0) at room temperature. In this case, ε-martensite embryos can be formed spontaneously, as shown in below images taken from Ramirez’s work published in Acta in 2016. Please bear in mind that, accumulation of regularly overlapped stacking faults form the ε martensite phase, which originates when the stacking sequence of close-packed (111)γ planes changes from ABCABC… to ABAB…, driven by the activation and motion of Shockley partial dislocations, as shown in Figure 4.
Q5. According to A5 - authors use in the answer the sentence "HRTEM" (i.e. high-resolution image). In the common use, the mentioned HRTEM uses when we can recognise much more details of structure e.g. atomic planes.
A5. Thanks for your comment. Changed to “The bright field TEM….”.
Q6. According to 9.2. It worth to note the recrystallisation process definitely needs some energy which we introduce into the material by plastic working. The recrystallisation the process which depends on a lot of factors. e.g. according to Boczwar relation, in the pure metals, the recrystallisation processes will start more quickly than alloys counterparts. The energy of stacking fault of material is smaller the more the stacking fault occurs in it. As a consequence, changing the slip plane (cross slip) is more difficult and, as a result, the material strengthens more. Therefore sentence "high storage energy is needed for recrystallization in low-stacking fault energy alloys" is very discussion. The present sentence suggests that high -stacking fault energy alloys need low storage energy - but .... isn't true. The materials with high-stacking fault strengthening much less than low-stacking fault energy materials.
A6. Perhaps it is a misunderstanding. The distance between partial dislocations in a material with low stacking fault energies is larger than a material with high stacking faults energies and this makes the material harder to deform plastically. And you are right, the cross-slip of dislocations is becoming more difficult, strengthening the material at the expense of decreasing the ductility.
Q7. According to A9 Thank you very much for the explanation. Unfortunately the sentences from the answer:"...images showing the precipitation of carbides on the stacking fault defects introduced during cyclic quenching" don't agree with the text in the manuscript: (line 263-267): "The other possibility to explain the grain refinement mechanism is the nucleation of the reversed austenite (γfcc) on the athermal ε martensite phase which is heavily faulted and can provide the potential nucleation sites for the new fcc grains and lead to the grain refinement near the surface layer, as shown in SEM micrographs in Figs. 10 c-d" Authors have to change it.
A7. This comment is not quite clear for us. The precipitation of carbides on the stacking faults (i.e. ε martensite) took place as shown in SEM images in Figures 10c and d. This has nothing to do with the mechanism of grain refinement which is either (1) recrystallization, or (2) formation of reversed austenite. If the reviewer asks the question more clearly, we will endeavour to address the comment.

Reviewer 3 Report
The paper has been revised accepting most of the reviewer's comments.
Author Response
Q1. The paper has been revised accepting most of the reviewer's comments.
A1. Thanks for allocating your time to review this paper.
Round 3
Reviewer 1 Report
Thank you very much for the changes in the manuscript and the "Explanations letter". The current version of the manuscript has the points that ought to be corrected:
According to Q1. Some part of the editing mistakes still exist but I think authors will correct on the publication process.
According to A3. "The X-ray map of carbon has been added" Where we can find it?
According to A4. Of course, the findings from present literature are very valuable but if we point out even similar phenomena we have to insert a reference to the papers which we used to make the findings.
"... Please bear in mind that, accumulation of regularly overlapped stacking faults form the ε martensite phase, which originates when the stacking sequence of close-packed (111)γ planes changes from ABCABC… to ABAB…, driven by the activation and motion of Shockley partial dislocations, as shown in Figure 4."
Can authors point out the mentioned structure changes in Fig.4 (with details)? In my opinion, it is hard to recognise the dislocation structure.
According to A6
I think it is a misunderstanding, too.
According to answer authors :
"...And you are right, the cross-slip of dislocations is becoming more difficult, strengthening the material at the expense of decreasing the ductility." - strengthening process leads to store more internal energy which is "driving motor" of mentioned recrystallisation.
At first, The stacking fault energies are an important characteristic and play a significant role in the deformation of metals due to its influence on dislocation mobility and morphology.
The recrystallisation process depends on not only stacking fault energy (!) but also: meting point, structure condition (pure chemical elements or alloys) grain refinement, condition of deformation process etc. Structures inhomogeneity is very important too (i.e. the existence of second phases or existence of a solid solution).
While materials with low stacking-fault energy can store a high portion of the internal elastic strains owing to the limited cross-slip and climb capabilities of the dislocations during recovery, alloys and metals and that have a high stacking fault energy such as aluminium can substantially reduce the deformation energy and hence also the remaining driving force for primary recrystallization during the preceding recovery period. In extreme cases, recovery can even suppress recrystallization.
Therefore the sentence :
"However, recrystallization in materials with low stacking fault energies (such as Co-Cr-Mo alloy) does not usually occur as easy as materials with high stacking fault energy (such as aluminium and aluminium alloys). (line 263-264)
...is not true and authors have to change it.
It worth to note that in materials with high stacking fault energy, such as aluminium, dynamic recovery can continuously remove and balance the strain hardening imposed during the hot working process.
Author Response
"However, recrystallization in materials with low stacking fault energies (such as Co-Cr-Mo alloy) does not usually occur as easy as materials with high stacking fault energy (such as aluminium and aluminium alloys). (line 263-264) ...is not true and authors have to change it. It is worth noting that in materials with high stacking fault energy, such as aluminium, dynamic recovery can continuously remove and balance the strain hardening imposed during the hot working process.
A4. Thanks for your comment. We have changed the sentence as requested, given below:
“In metals with a low value of γSFE, the difficulty of cross slip and/or climbing of dislocations reduces the ability of the material to accommodate plastic deformation by slip alone (basic mechanisms responsible for recovery), and therefore deformation twinning may occur. The development of subgrains are not usually seen in metals with lower stacking fault energy such as stainless steel and Cobalt-base alloys, because recrystallization usually occurs prior to significant recovery.”
Just to be clear:
The stacking fault energy, which is related to the atomic bonding in the material, determines the extent to which unit dislocations dissociate into partial dislocations. Such dissociation, which is promoted by a low value of γSFE, hinders the climb and cross slip of dislocations, which are the basic mechanisms responsible for recovery. Dislocation theory therefore predicts that high values of γSFE should promote dynamic dislocation recovery. In metals with a low value of γSFE, the difficulty of cross slip reduces the ability of the material to change its shape during plastic deformation by slip alone, and therefore deformation twinning may occur. Well-developed subgrain structures are not usually observed in metals of lower stacking fault energy such as stainless steel, because recrystallization occurs before significant recovery can occur. However, in materials of lower stacking fault energy such as nickel and stainless steel, recovery is slow, and the dislocation density increases to the critical value necessary for dynamic recrystallization to occur.
